

# Multidecadal Variations of the Effects of the Quasi-Biennial Oscillation on the Climate System

Stefan Brönnimann[1,2], Abdul Malik[1,2], Alexander Stickler[1,2], Martin Wegmann[1,2], Christoph C.
Raible[1,3], Stefan Muthers[1,3,4], Julien Anet[5], Eugene Rozanov[6,7], Werner Schmutz[7]

[1] Oeschger Centre for Climate Change Research, University of Bern, CH-3012, Switzerland
[2] Institute of Geography, University of Bern, CH-3012, Switzerland
[3] Climate and Environmental Physics, Physics Institute, University of Bern, CH-3012, Switzerland
[4] now at: German Meteorological Service, Research Center Human Biometeorology, Freiburg, D-79104, Germany
[5] Swiss Federal Laboratories for Materials Science and Technology, Dübendorf, CH-8600, Switzerland
[6] Institute of Atmospheric and Climate Sciences, ETH Zurich, CH-8092, Switzerland
[7] PMOD/WRC Davos, CH-6260 Davos, Switzerland

*Correspondence to*: Stefan Brönnimann (stefan.broennimann@giub.unibe.ch)

**Abstract.** Effects of the Quasi-Biennial Oscillation (QBO) on tropospheric climate are relatively small or appear only intermittently. Studying them requires long time series of both the QBO and climate variables, which has restricted previous studies to the past 30-50 years. Here we use the benefits of an existing QBO reconstruction back to 1908. We first investigate additional, newly digitized historical observations of stratospheric winds to test the reconstruction. Then we use
the QBO time series to analyze atmospheric data sets (reconstructions and reanalyses) as well as the results of coupled ocean-atmosphere-chemistry climate model simulations that were forced with the reconstructed QBO. We investigate effects related to (1) tropical-extratropical interaction in the stratosphere, wave-mean flow interaction, and subsequent downward propagation and (2) interaction between deep tropical convection and stratospheric flow. We generally find weak connections, though some are statistically significant over the 100-year period and consistent with model results. Apparent
multidecadal variations in the connection between the QBO and the investigated climate responses are consistent with a small effect in the presence of large variability, with one exception: the imprint on the northern polar vortex, which is seen in recent reanalysis data, is not found in the period 1908-1957. Conversely, an imprint in Berlin surface air temperature is only found in 1908-1957, but not in the recent period. In the model simulations, likewise, both links tend to appear alternatingly, suggesting a more systematic modulation. Over the Pacific warm pool, we find increased convection during easterly QBO
mainly in boreal winter in observation-based data as well as in the model simulations, with large variability. No QBO effects were found in the Indian monsoon strength or Atlantic hurricane frequency.





## 1 Introduction

The Quasi-Biennial Oscillation (QBO) is an oscillation of equatorial stratospheric zonal winds with a downward propagating phase taking approximately one year from the stratopause to the tropopause. It is relevant for interannual variability of stratospheric dynamics and composition (Baldwin et al., 2001), both in the tropics and the polar regions (e.g., Holton and

Tan, 1980). It has also been demonstrated that the QBO affects tropospheric weather, either through its effect on the stratospheric polar vortex (Baldwin et al., 2001) or perhaps directly through interaction with tropical convection (Collimore et al., 2003; Huang et al., 2012). Tropospheric imprints were found in the Eurasian region, including the North Atlantic or Arctic Oscillation and Eurasian snow cover (e.g., Peings et al., 2013). The QBO has also been claimed to affect the Indian monsoon system (e.g., Mukherjee et al., 1985), Atlantic hurricane frequency (Klotzbach, 2007) or El Niño/Southern

Oscillation (Gray et al., 1992a,b). Given the QBO's close-to-periodic variation (which implies predictability), any such mechanism raises hope to increase the prediction skill beyond the classical weather forecast of a couple of days (see Tripathi et al., 2015). Furthermore, the QBO might modulate forcing-response relationships. For instance, Labitzke et al. (2006) found that the QBO modulates the effect of solar activity on the polar vortex (see also Roy and Haigh, 2011).

However, the effects of the QBO on the troposphere are rather weak and some appear only intermittently, e.g. between the

1950s and the 1980s (e.g., Camargo and Sobel, 2010). This calls for an analysis of long time series. The standard QBO time series - that of the Freie Universität Berlin (FUB) - reaches back to 1953. Spectral analyses from earlier data confirm that the QBO existed before that time (Labitzke and van Loon, 1999). Schove (1969) and others analysed historical observations for evidence of the QBO. In fact, a clear 2.2-yr cycle in Berlin surface air temperature (SAT) was already reported by Baur (1927). Landsberg (1962) and Landsberg et al. (1963) used spectral analysis of many more SAT series and found 2.2 yr

cycles. Spectral analyses of the North Atlantic Oscillation or the Northern Annular Mode indices also indicate significant peaks near 2.2 years (Hurrell and van Loon, 1997; Coughlin and Tung, 2001). More generally, speculations of a biennial oscillation in climate variables go back to the 19[th] century (see Clayton (1884, 1885) for precipitation and pressure in the United States, Woeikof (1895) for Scandinavian snow cover). However, for the period prior to 1953, no direct comparison of stratospheric wind data and independent tropospheric climate data could so far be done, which is the aim of our paper.

We base our analysis on a previous paper (Brönnimann et al., 2007), where we have attempted to reconstruct the QBO back to 1908 by using the solar semidiurnal tide extracted from hourly sea-level pressure (SLP) data. The reconstruction was augmented by incorporating historical stratospheric wind observations and was validated using the QBO signature in historical total column ozone data. In the meantime, we have digitised a large amount of additional historical upper-air data (Stickler et al., 2014a), which partly cover the equatorial stratosphere. Among the new data are Berson's 1908 observations

in East Africa (Brönnimann and Stickler, 2013), wind profiles from Batavia, 1910-1911, profiles from cruises of the research vessel Meteor 1925-1927 (Stickler et al., 2015), as well as several other measurements. We first present the new data sources and compare the results to the previous reconstruction. After finding that the additional data do not contradict the previous reconstruction, we use the reconstruction to analyse the imprint of the QBO in observations-based data sets (historical




reanalyses and reconstructions) of tropospheric circulation and climate. We analyse SAT, precipitation, snow cover, tropospheric wind fields, and hurricane tracks. The same analyses are performed in a set of four simulations with a coupled ocean-atmosphere-chemistry-climate model that were nudged to the same reconstructed QBO, but backwards extended to 1600, such that we have 4 × 405 years of model data available (Muthers et al., 2014).

The paper is organised as follows. In Section 2, we describe the historical upper-air data and the quality check of the QBO reconstruction and describe the climate model simulations. Results are presented in Section 3 and discussed in Section 4. Conclusions are drawn in Section 5.

## 2 Data and Methods

### 2.1 QBO Time Series

**2.1.1 Historical Evidence**

Only sporadic information is available on the QBO before 1953. The first indirect indications of stratospheric wind variability relate to observations of volcanic plumes (Hamilton, 2012). The most famous example is the observation of the Krakatau volcanic plume in 1883, which circled the globe from east to west. The high altitude winds (the stratosphere was not yet discovered) responsible for this transport became widely known as „Krakatau easterlies". Direct observations of
equatorial stratospheric winds by means of balloons go back to 1908, when Berson, in an expedition to East Africa, reported unexpected westerly winds in the lower stratosphere (Süring, 1910). These westerlies were confirmed by van Bemmelen and Braak (1910), who performed observations of upper-level winds in Batavia from 1909 to 1918. Lower stratospheric westerlies were also confirmed by the observations of another volcanic eruption plume (Semeru, 15 Nov 1911) as reported by Hann and Süring (Hamilton, 2012). Reconciling Berson's westerlies with the expected easterly winds remained a
challenge until the discovery of the QBO in the 1960s (Hastenrath, 2007).

Direct observations of equatorial stratospheric winds were very sparse prior to the 1950s. Results were summarized by Schove (1969) and Hamilton (1998). After the 1950s, when a global radiosonde network was built up, stratospheric winds were operationally observed in the equatorial region. It was in these data that Reed et al. (1961) and Veryard and Ebdon (1961) discovered the QBO. Based on radiosonde records from Canton Island (3° S, 172° W) from 1953 to 1967, Gan
(Maledives, 1° S, 73° E) from 1967 to 1975 and Singapore (1° N, 104° E) since 1976, Naujokat (1986) and Marquardt and Naujokat (1997) were able to derive the QBO time series back to 1953, known as the FUB QBO. Since the advent of reanalysis data, the QBO is normally defined as the zonally averaged zonal wind in the stratosphere at the equator. We follow this definition.





### 2.1.2 Reconstruction of the QBO

In this paper we use the monthly reconstruction of the QBO (zonal mean zonal winds at the equator) from Brönnimann et al. (2007). This reconstruction is based on the surface signature of the QBO-modulated solar semidiurnal tide in hourly surface pressure observations from Batavia prior to 1945 as well as on historical upper-air wind profiles. For the reconstruction we
first defined a perpetually repeating "ideal QBO cycle" from deseasonalized reanalysis data. Then we used the observational evidence to determine a time axis (i.e., timing of phases) and interpolated the ideal cycle onto this new time axis. Finally, we added back the annual cycle. We used historical total ozone data (which also show an imprint of the QBO) to assess the reconstruction and found generally good agreement, but the real QBO might be out of phase by up to 3 or 4 months.
The reconstruction is supported by historical upper-air observations mainly in the 1910s and in the 1940s, while the solar
semidiurnal tide provides continuous information but stops in 1945, afterwards the reconstructions are entirely based on upper-air wind observations. The Freie Universität Berlin QBO starts in 1953. From September 1957 on, the QBO is taken from ERA-40 and after 1979 from ERA-Interim (Dee et al. 2011). The resulting 108-year QBO record is given in Fig. 1. We are currently in the 48[th] cycle since Berson's profile, which we take as a starting point of our work. The number of cycles thus allows robust statistics.

### 2.1.3 Additional Historical Upper-Air Data

The data presented here are part of a set of 1.25 million upper-air profiles that were digitised in the framework of the ERA-CLIM project (Stickler et al., 2014a,b), adding to the 12.75 million upper-air winds profiles that were already available from the CHUAN data compilation prior to 1957 (Stickler et al., 2010). A plot of most of the equatorial stratospheric data comprised in the latter data set was already given in Labitzke et al. (2006) and they entered the reconstruction described
above. For this paper we collected all additional (ERA-CLIM) data prior to 1950 from stations within 20° S to 20° N. In the following we highlight three particular records.

*Berson's East Africa Expedition*
In 1908, the German meteorologist Arthur Berson organized an aerological expedition to East Africa with the initial aim to better understand the upper-level branch of the trade wind and monsoon systems (Fig. S1 shows the launch of a registering
balloon on Lake Victoria). Upper-level winds were observed with pilot balloons and registering balloons (briefly described in Brönnimann and Stickler, 2013). Only few wind profiles reached the stratosphere. Surprisingly, some of them indicated westerly winds in the stratosphere. Figure S1 (right) shows the wind profiles that reached the stratosphere. Although all profiles except two were taken during an only 15-day interval, there is considerable scatter. It is very difficult to identify wind regimes from the raw data directly, although there are westerly winds in the stratosphere in several profiles.
The corresponding profiles from the reconstructions are also indicated. While there is a good agreement with some of the profiles (westerlies between 18 km and 20 km and easterlies above), others show relatively strong easterlies between 16 and



18 km (or even higher), where the reconstructions suggest zero zonal wind. Note that the reconstructions assumed westerly winds at 19 km altitude throughout the year 1908 based on the notion of Berson westerlies.

*The Batavia Data*

In 1909 the Dutch Colonial Secretary started aerological observations in Batavia (van Bemmelen et al., 1911). Leading
aerologists such as Richard Assmann and Hugo Hergesell advised the scientists, and measurements could be started in the course of 1909. Kites, registering balloons and pilot balloons were used. Many of the balloons reached high altitudes, and soon westerlies winds were observed (van Bemmelen and Braak, 1910), thus confirming Berson's findings. Veryard and Ebdon (1961) and Ebdon (1963) analyzed the Batavia winds from 1909 to 1918 (which they published in the form of monthly averaged wind directions for certain altitude bands) and found a clear QBO signature, including the downward
phase propagation. Their published phases are interpretations, not raw data, and these phases were used to constrain our reconstructions. From the digitized data (Fig. S2 shows the earliest phase of measurements) it is however difficult to discern clear wind regimes.

*Research vessel „Meteor"*

From 1925 to 1927, the German research vessel Meteor cruised the Atlantic and took, in addition to many oceanographical
measurements, also aerological observations. More than 1000 vertical profiles were retrieved on east-west transects across the tropical and South Atlantic between 20°N and 64° S (see Stickler et al., 2015 for details). Apart from kites and some registering balloon ascents (none of which reached the stratosphere), 801 pilot balloon ascents are available, of which the highest reached 20.5 km. In the tropical region (20°N to 20°S), however, only few measurements higher than 16 km are available.
These measurements (as well as those from Berson, Batavia and all other measurements from the ERA-CLIM data set) are incorporated as circles into Figure 1. Again it is difficult to find a coherent picture. Counting the agreement in sign of the zonal wind between observations and reconstructions, we find an agreement of 60%. This is not a particularly good score for an evaluation, which is not even fully independent. The rate increases, though, if we only use observations above 19 or 20 km or exclude comparisons for which reconstructed winds are weak (i.e., close to phase change). Conversely, there is no
systematic pattern of disagreement (no out-of-phase relation). We therefore continue and use our reconstructions for further analyses but note that the reconstruction remains to be further confirmed.

**2.2 Tropospheric Circulation and Climate Data**

In order to analyse the imprint of the QBO in historical times, we use several data sets of the tropospheric circulation that cover the pre-1957 period. These data sets include the Twentieth Century Reanalysis (20CR; Compo et al., 2011), versions 2
and 2c, the reanalysis ERA-20C Deterministic (Poli et al., 2016) as well as several data sets based on reconstructions. As a reference in the recent period we use the ERA-Interim reanalysis.


20CR is based on the assimilation of surface or SLP from the International Surface Pressure Data Bank (ISPD) and the International Comprehensive Ocean-Atmosphere Data Set (ICOADS), with monthly sea-surface temperatures (SSTs) and sea ice used as a boundary condition. Versions 2 and 2c differ with respect to the ISPD versions used (v2 and v3.9, respectively), the starting year (1871 and 1851) and the SSTs used (HadISST1.1; Rayner et al., 2003, and SODAsi version 2;

Giese et al., 2015, with the high latitudes (>60°) corrected to COBE-SST2; Hirahara et al., 2014). Previous validation studies have shown that 20CR agrees well with independent observations in the midlatitudes, but less so in the tropics. In this study we use the ensemble mean monthly mean data.

The ERA-20C reanalysis reaches back to 1900 and uses very similar pressure input as 20CRv2c, but additionally also assimilates marine winds and uses a newer version of HadISST (HadISST2; see Poli et al., 2016). Since the results were very

similar, we show only the results from 20CRv2c for most of the analyses.

In addition to 20CR we use monthly mean fields of wind and geopotential height (GPH) at different levels from a statistical reconstruction (Griesser et al., 2010), which reaches back to 1880. It is based on historical upper-air (after 1918) and surface data, which were calibrated against ERA-40 (Uppala et al., 2005) in a principal component regression approach. Here we use GPH at 100 hPa in the northern extratropics.

Further we also use monthly indices that were reconstructed based on surface and upper-level variables using regression approaches (Brönnimann et al., 2009; Zhou et al., 2009) calibrated against (and extended by) NCEP/NCAR Reanalysis (Kistler et al., 2001). The same indices were also calculated from 20CRv2c. Specifically, we use the indices Z100, defined as GPH difference between 75-90° N and 40-55° N at 100 hPa (Brönnimann et al., 2009) as a measure for the weakness of the polar vortex, the Pacific Walker Circulation index PWC (the difference in vertical velocity at 500 hPa between the areas

[10°S–10°N, 180–100°W] and [10°S–10°N, 100–150°E] following Oort and Yienger (1996)), and the Dynamic Indian Monsoon Index DIMI (the difference in 850 hPa zonal wind between the areas [5–15°N, 40–80°E] and [20–30°N, 70–90°E] following Wang et al. 2001).

We further used GHCNv3 SAT from Berlin, a time series of Atlantic hurricane activity (Vecchi and Knutson, 2011) as well as the HadCRUT4v global SAT data set (Morice et al., 2012).

## 2.3 Climate Model Simulations

The reconstructed QBO, backward extended to 1600 by repeating the ideal QBO cycle plus seasonal cycle, was used to nudge the coupled chemistry-climate-ocean-atmosphere model SOCOL-MPIOM. The simulations are described in Muthers et al. (2014). In brief, SOCOL-MPIOM is a combination of the chemistry-climate model SOCOL version 3 (Stenke et al., 2013), which is based on the middle atmosphere version of ECHAM5 (Roeckner et al., 2006; Manzini et al., 2006) coupled

to the chemical module MEZON (Model for Evaluation of oZONe trends (Rozanov et al., 1999; Egorova et al., 2003), and the ocean model MPIOM (Marsland, 2003; Jungclaus et al., 2006). The atmospheric model was run at a resolution of T31 (approx. 3.75° × 3.75°), with 39 levels (model top at 0.01 hPa / 80 km). The ocean state in 1600 was branched off another simulation (Jungclaus et al., 2010; see Muthers et al., 2014; Anet et al., 2014; for more details).





Four simulations were performed for the period 1600 to 2000. Two simulations, termed F13 and F14 (differing only in their initial state) use a relatively strong solar forcing while the two simulations F23 and F24 (again differing only in their initial state) use a weaker solar forcing. As will be shown later, no differences between the ensemble members are found with respect to their QBO effects. We thus also analysed a sample in which the four simulations were pooled.

**2.4 Methods**

The following methodology is applied to all analyses. Target variables and fields are analysed on a seasonal scale. Therefore, we first defined seasons that pertain to easterly or westerly phases of the QBO. For testing those hypotheses that require interaction with the stratospheric polar vortex and downward propagation, we defined the QBO phase from 50 hPa tropical zonal mean wind in early winter (Nov.-Dec.) and then compared the climate records of the late winter (Jan.-Mar; March for

snow cover). Months close to the reconstructed phase change (which is uncertain by up to 3-4 months) are excluded by requiring tropical zonal mean wind at 50 hPa to exceed 5 m/s over a two months period. We chose these seasons because interaction between the QBO and the polar vortex may occur in early winter. The downward propagation from the polar stratosphere to the surface than takes several weeks and the signal may persist in the troposphere.

For those hypotheses that involve direct interaction between the QBO and high-reaching convection, we used the 70 hPa

QBO in boreal summer (JJ), and winter (ND) and analysed the climate fields over a three months period starting with one month offset (JAS, DJF, respectively). The JAS definition was preferred over the more classical JJA period following Chattopadhyay and Bhatla (2002), who found a stronger QBO signature in the Indian Monsoon in that season. The shorter lag (as compared to the polar vortex based analyses described above) allows a more direct adjustment of the tropospheric circulation to stratospheric forcing. Periods when tropical zonal mean wind was weaker than 3 m/s were not considered.

Furthermore, for all analyses we excluded years following major tropical volcanic eruptions (i.e., 1601, 1642, 1675, 1720, 1730, 1810, 1816, 1832, 1836, 1884, 1912, 1926, 1964, 1983, and 1992).

Note that for specific problems, more accurate definitions of seasons could be found (see Gray et al., 1992a,b; Huang et al., 2012) at the price of simplicity or (possibly) independence. When defined in the above way, we find 39 easterly and 43 westerly phases for the boreal winter 50 hPa QBO since 1908. The corresponding numbers for the 70 hPa QBO for summer

(winter) are 26 (31) easterly and 36 (30) westerly phases. The number of phases in each of the model simulations is about four times larger.

Our main method is a composite analysis of the two phases (easterly minus westerly), using heteroscedastic t-testing to assess statistical significance. In the observation-based data we do this for the entire time period as well as for the subperiods 1908-1957 and 1958-2012. As a reference we also apply the composite analysis to 1979-2015 in ERA-Interim data. In the

model we apply the method to the entire time period for all simulations separately as well as for the pooled simulations. In the paper we show composite fields only for the latter; corresponding composites for all ensemble members and variables, including statistical significance, are shown in the Supplement. Finally, we also performed 30-yr moving composites, both in





the observation-based data and in the model (only in the individual members). We then calculated standardized differences, using the standard deviation over the entire period.

## 3. Results

### 3.1. QBO-Polar Vortex Interaction and Downward Propagation

#### 3.1.1 The QBO Effect on the Polar Vortex and the NAO

Several mechanisms responsible for QBO influence on tropospheric climate have been proposed. One pathway, known as the Holton-Tan effect (Holton and Tan, 1980; Baldwin et al., 2001), is through the extratropical stratosphere in boreal winter. This mechanism is understood to operate via changes in the extratropical planetary wave activity flux. An easterly QBO at 50 hPa leads to convergence of wave activity in the subtropical lower stratosphere and in subpolar middle and upper

stratosphere (e.g., Garfinkel et al., 2012). The waves deposit easterly momentum and decelerate the mean flow. The signal can propagate downward and eventually reach the Earth's surface, although the mechanism is still not fully understood (see Anstey and Shepherd, 2014, Kidston et al. 2015; a review of the proposed mechanisms is beyond the scope of this paper).

Compositing easterly minus westerly QBO in boreal winter in ERA-Interim (Fig. 2) shows this classical response. The zonal mean zonal wind weakens, most strongly at around 30 km, 70 °N. Cooling is found above and warming below. GPH exhibits

positive anomalies poleward of 60 °N in the lower stratosphere, indicative of a weak polar vortex.

Reconstructions and reanalyses both do not provide information for altitudes above around 10-15 km. The highest level we analyse here is 100 hPa (note that this might be too low to capture QBO effects but too high to be well reconstructed). In the Z100 index (Table 1) we find no significant effect in any of the subperiods. The early period even shows a negative difference (thus opposite to what is expected from Fig. 2). A more consistent relation is found within the model simulations,

which is highly significant for the pooled sample and is significant at the 95% (90%) level for 2 (3) out of the 4 simulations (not shown).

Compositing 100 hPa GPH and 200 hPa zonal wind for January to March gives similar results (Fig. 3). The analysis of 20CRv2c (1908-2012) and of the simulations (ensemble mean) show almost opposite patterns, but it should be noted that there is hardly any significance in the 20CRv2c composites (Fig. S3; no significance at all is found in 100 hPa GPH; not

shown). In order to test whether uncertainties in 20CRv2c in the early times could be the cause of that, we compared the composite for 100 hPa GPH for the 1908-1957 period between 20CRv2c, ERA20C, and statistical reconstructions (Fig. S4). In fact, there are some differences between the products. 20CRv2c exhibits a stronger negative signal over the polar region than the other data sets, but none shows the weakening of the vortex expected from the Holton-Tan effect.

The difference in the QBO imprint between the individual model simulations (Fig. S6) is smaller than between model and

20CRv2c analyses. Each of model simulations shows a significant signature over the polar region as well as between 35° and



45° N. The weakening of the zonal wind found in ERA-Interim in the recent period is reproduced qualitatively in the model simulations, but not in 20CRv2c.

The lack of a consistent signal in two subperiods could point to the lack of a signal in general or to an intermittent behaviour of the QBO signature. The model results suggest that the signal might be small (though significant), such that short periods
may by chance show an opposite relation. To test this, East-West differences in 30-yr moving windows are analysed (Fig. 4). In fact, the standardized differences for such periods vary considerably in the model. In the observation-based data, the difference was largest for the interval 1960 to 1989, i.e., close to the time window in which the Holton Tan effect was originally discovered.

In all, the reconstructed QBO and polar vortex strength at 100 hPa from reconstructions and reanalysis prior to 1957 do not
show a relation. This could be due to inferior data quality of either or a too low analysis level. The model does show a significant relation as expected from the Holton-Tan effect, but the signal is rather small or transient. Results are consistent with ERA-Interim considering decadal variability as found in the model.

To assess whether the QBO affects the surface the ERA-Interim analysis (Fig. 2) is again consulted. Zonal averages indeed show small surface effects (weaker zonal wind, higher SAT and pressure for easterly phases), but only poleward of 80 °N.
Very often, the NAO index is analysed as an indication for surface imprints of stratospheric perturbations. This index is well constrained in 20CR (Compo et al. 2011) and hence the NAO index is treated similarly as Z100. As expected, differences in the NAO have the opposite sign as those for Z100. However, none of the differences in 20CRv2c are significant. In the model simulations, differences are significant at the 90% level in one out of four simulations (the p-value for the pooled sample is exactly 0.05). The 30-yr moving window composite of the NAO index shows that the decadal variability of the
difference is large (both in observations and model simulations), but anti-correlated with that of Z100 (which is expected).

Perhaps the simple dipole-NAO index is not suitable to capture the response. We therefore also analysed composites of SLP fields. We find negative anomalies at midlatitudes stretching from the eastern North Atlantic to central Eurasia in both 20CRv2c and model simulations (a similar pattern was recently found by Roy et al. 2016). Indeed, the pattern is shifted southward as compared to a classical NAO pattern. In the North Atlantic-European region, the agreement between model
and observation-based data is stronger at the surface than in the stratosphere (note that the pattern over North America, in contrast, is almost opposite in 20CRv2c and model simulations). In the model, the signature is consistent in all four simulations (Fig. S5). Thus, both the signatures in the stratosphere and in SLP are consistent with the Holton-Tan effect, but the variability is so large that even with very long records, results remain near the limit of significance.

### 3.1.2. The QBO Effect on Berlin Surface Air Temperature

Baur (1927) analysed the 100-yr record of Berlin SAT and found a very clear quasi-biennial cycle. Within our QBO reconstructions, we also find highly significant differences for Berlin SAT (we used observations rather than reanalyses) in winter (Jan.-Mar.) between East and West phases of the QBO (50 hPa, Nov.-Dec.). SAT is higher during easterly phases of the QBO. This is unexpected as Berlin SAT is positively correlated with the NAO and negatively with Z100. The effect





might be real as the model simulations (grid point 15° E, 50.1° N) also show higher SAT during the easterly phase of the QBO as compared to the westerly phase, albeit not significantly.

Interestingly, the difference is significant only in the first period (which is when Baur (1927) discovered 2.2-yr cyclicities in Berlin SAT) and over the entire period, but not in the post-1957 period. In other words, the difference was significant in the period when no effect in Z100 and NAO was found. The 30-yr moving windows difference in the model simulations shows a similar behaviour. There are multidecadal periods when the QBO signature in Berlin SAT is positive while the NAO (Z100) signature is around zero, and periods when the NAO signature is negative and that in Berlin SAT is around zero. As for the raw series, the 30-yr moving windows difference series of Berlin SAT, -1 × Z100, and NAO are positively correlated (numbers in Fig. 4), although over the entire period, the sign is opposite (most circles are above or to the left of the one-to-one line). This suggests that part of the decadal variability in the QBO-surface climate relation might arise from decadal climatic variability such as latitudinal shifts of circulation features.

The composite field of SAT based on 20CRv2c (HadCRUT4v shows similar results) reveals that the warming for easterly phases stretches across much of Eurasia. It seems well reproduced in the model simulations, where it maximizes between the Caspian and Aral Seas. However, there is quite a large discrepancy between individual simulations despite the fact that they are 405 years long (Fig. S7). The SAT signal over North America is totally different.

Based on these results, we defined a new SAT index for the Caspian-Aral Sea region, which is the region with the strongest imprint in the model composites. Even in this optimized case, there are some (albeit few) 30-yr periods in the 4 × 405-yr model simulations that would exhibit a significantly negative relation when analysed in isolation. Interestingly, correlations between the 30-yr moving windows difference series of the new SAT index and those of NAO and -1 × Z100 are predominantly negative, as expected from the Holton-Tan effect.

### 3.1.3. The QBO Effect on Snow Cover

Woeikof (1895) speculated that snow cover follows a biennial cycle. To test this, snow depth in March is analysed. The corresponding composites (see Fig. 3) are highly consistent with the results for SAT, but again do not show a systematic effect. The high-resolution snow cover product from ERA-20C shows very similar results as 20CRv2c (see Fig. S5), i.e., the QBO East minus West differences for the two subperiods differ, and they both differ from the model simulations.

From these analyses there is no indication that snow cover in March is affected by the QBO in a significant way. Conversely, we can also not exclude intermittent effects. Peings et al. (2013) found an effect of the QBO on Siberian snow cover, but only after 1976 and not before. Hence, Woeikof (1895) might still have captured a QBO signal when finding differences in snow cover in Scandinavian between even and odd years – or (what is more likely) he was picking up random variability.





### 3.2. QBO Interaction With Deep Tropical Convection

### 3.2.1. The QBO Effect on the ENSO System and the Pacific Walker Circulation

In 1992, Gray et al. (1992a,b) suggested an effect of the QBO on the ENSO system. Later publications addressed the effect of the QBO on tropical convection in observations (e.g., Collimore et al., 2003; Huang et al., 2012; Liess and Geller, 2012)

or models (e.g., Giorgetta et al., 1999; Garfinkel and Hartmann, 2011; Nie and Sobel, 2015). Several mechanisms have been suggested as to how a link might proceed. Giorgetta et al. (1999) found that wind shear near the tropopause associated to the QBO phase in the lowermost stratosphere affects deep convection in the warm pool area. Huang et al. (2012) argued that the change in static stability due to the temperature QBO might play a more important role. However, the role of clouds and other feedbacks is not well understood (e.g., Garfinkel and Hartmann, 2011).

According to the wind shear mechanism, lower shear would favour deeper convection. Climatologically, easterlies dominate over the warm pool in the uppermost troposphere, hence an easterly QBO phase at 70 hPa reduces the wind shear and would enhance convection. With respect to the temperature, a westerly QBO phase at 70 hPa is associated with warm layer below, leading to increased stability in the tropopause region and thus less convection. From both mechanisms we expect more convection during easterly phases of the QBO in the lower stratosphere. Therefore, stability and wind shear influences

cannot easily be separated without more detailed diagnostics that are not available for our study.

The analysis in ERA-Interim (Fig. 5) for zonal averages over Indonesian and the Pacific Warm Pool (120° E to 160° E) in boreal winter is consistent with the suggested mechanism. While SAT and zonal wind do not show a tropospheric imprint, an increase in tropical convection is found for easterly phases. This increase is shifted towards the northern hemisphere relative to the climatological maximum in convection.

As a first approach, we analysed the PWC index, which is well reconstructed and rather well constrained in 20CR (see Compo et al., 2011). We found no significant signature in observation-based data, but in the model the Walker circulation is stronger for easterly than for westerly QBO in both seasons (JAS and DJF), in some simulations highly significant. This is consistent with increased convection over the Pacific warm pool area (Fig. 5).

This imprint can be better understood when analyzing fields rather than an index (although the fields are less reliable).

Composites of SST, vertical velocity, and zonal winds at 10 m and at 200 hPa are shown in Figure 6 (see Fig. S10 for significance). The most obvious signature is a slight eastward shift of the centre of convection over Indonesia during easterly QBO phases, resulting in the pattern seen in ERA-Interim. This is seen in both seasons (though stronger in boreal winter) and it is also seen in observations. The response thus does not project well onto the Pacific Walker circulation (only significant in the model) and surface winds over the central Pacific remain unaffected. Signatures in SST show a slight

equatorial Pacific warming but a cooling in Indonesia. However, for these findings significance is also limited (see Fig. S8 and S9). The winter hemisphere subtropical jet moves poleward in 20CRv2c and to some extent also in the model simulations.





### 3.2.2. QBO Effect on Atlantic Hurricanes and on the Indian Summer Monsoon

Finally, we also briefly analyse the relation between the QBO and Atlantic hurricanes or the Indian Summer monsoon strength. In both cases, our results revealed no significant differences with the simple indices used. Note, however, that for the Indian summer monsoon the relation might be more complex (e.g., Claud and Terray, 2007). The existence of a

Tropospheric Biennial Oscillation in the summer monsoon has been claimed (Meehl et al., 2003), but this might arise from white noise and not from deterministic processes (Stuecker et al., 2015).

### 4. Discussion and Conclusions

Our analysis reveals relatively small influences of the QBO on the tropospheric climate, which are however consistent with historical literature (which is not too surprising as the underlying observation-based data are partly the same) and in some

cases with climate model simulations. Although issues of data quality also contribute in the case of observation-based data, it is interesting that even with very long time series and very long model simulations, only few statistically significant results are found. Multidecadal variations of QBO-climate links are mostly consistent with a small signal in the presence of internal decadal climate variability, although one of the results (the fact that periods with a QBO signatures in Berlin surface temperature and in the NAO alternate) also points to possible climatic modulations.

Based on the analysis of 108 years of QBO and climate variables from reconstructions as well as $4 \times 405$ years of data from climate model simulations, we thus conclude that:

      (1) there is no evidence that the extended QBO reconstruction is out-of phase with the true QBO, but further support for the reconstructions is clearly required,

      (2) the relation between the QBO and climate variables is weak and characterized by large multidecadal fluctuations,

(3) in boreal winter, there are links between the QBO and the stratospheric polar vortex or between the QBO and Berlin SAT, but the former relation is typically strong when the latter is weak and vice versa (both in model and observations), suggesting a climatic origin of the decadal modulation, the relation to a more broadly defined Eurasian SAT index is more stable

      (4) there is a weak but significant effect of the QBO on deep convection over the Pacific Warm Pool, mainly in boreal

winter (an eastward shift of convection during easterly QBO in the lowermost stratosphere); though significant, this change does not project strongly onto ENSO

our results are consistent with historical literature and also with the sequence of discoveries of biennial imprints in weather and climate, as expected for analyses of small effects embedded within strong variability.




**Acknowledgements.**

This paper is dedicated to Karin Labitzke, who made major contributions to the understanding of QBO effects on climate. The work was supported by the Swiss National Science Foundation under grant CRSII2-147659 (FUPSOL II) and the EC FP7 project ERA-CLIM2.

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





**Tables**

Table 1. Standardised difference in indices between easterly and westerly QBO phase in different observation-based data sets as well as in the climate model simulations (F?? denote the simulation numbers, also shown is the result for the pooled simulations). Bold and italics denote significance at the 95% and 90% level, respectively.

| Index | 1908-2014 | 1908-1957 | 1957-2014 | F013 | F014 | F23 | F24 | Pooled |
|---|---|---|---|---|---|---|---|---|
| Z100 (REC+NNR) | 0.226 | -0.037 | 0.425 | **0.304** | *0.206* | **0.279** | 0.163 | **0.238** |
| Z100 (20CRv2c) | -0.242 | -0.443 | -0.123 | | | | | |
| NAO (20CRv2c) | 0.088 | 0.356 | -0.100 | -0.002 | *-0.213* | -0.183 | -0.081 | *-0.120* |
| Berlin Temp (GHCN) | **0.513** | **1.217** | 0.015 | *0.209* | -0.158 | 0.012 | 0.159 | 0.056 |
| PWC$_{DJF}$ (REC+NNR) | 0.049 | 0.004 | -0.003 | **0.280** | **0.340** | **0.367** | 0.026 | **0.253** |
| PWC$_{DJF}$ (20CRv2c) | 0.111 | 0.002 | 0.002 | | | | | |
| PWC$_{JAS}$ (REC+NNR) | *-0.497* | -0.381 | -0.676 | **0.356** | 0.108 | **0.235** | -0.127 | **0.143** |
| PWC$_{JAS}$ (20CRv2c) | -0.289 | -0.323 | -0.254 | | | | | |
| DIMI$_{JAS}$ (REC+NNR) | 0.164 | -0.239 | 0.713 | -0.044 | 0.040 | -0.091 | 0.005 | -0.022 |
| DIMI$_{JAS}$ (20CRv2c) | 0.250 | 0.108 | 0.509 | | | | | |
| Hurricanes | -0.100 | -0.016 | -0.173 | NA | NA | NA | NA | NA |





**Figure 1:** Hovmöller diagram (time–height cross section) of zonal-mean zonal wind at the equator from 1908 to 2015 (from Brönnimann et al., 2007). The dots indicate the additional wind data rescued within the ERA-CLIM project.




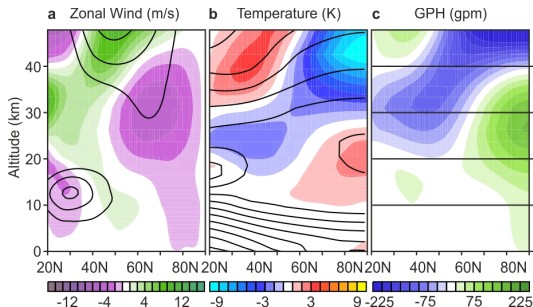

**Figure 2:** Composite fields (shading) and climatologies (contours) of ERA-Interim for boreal winter for easterly minus westerly QBO phases for zonal averages of (a) zonal wind (contours: 20 to 60 m/s in steps of 10 m/s), (b) temperature (contours: 200 to 300 K in steps of 10 K) and (c) GPH in Jan.-Mar. using the 50 hPa QBO early winter.



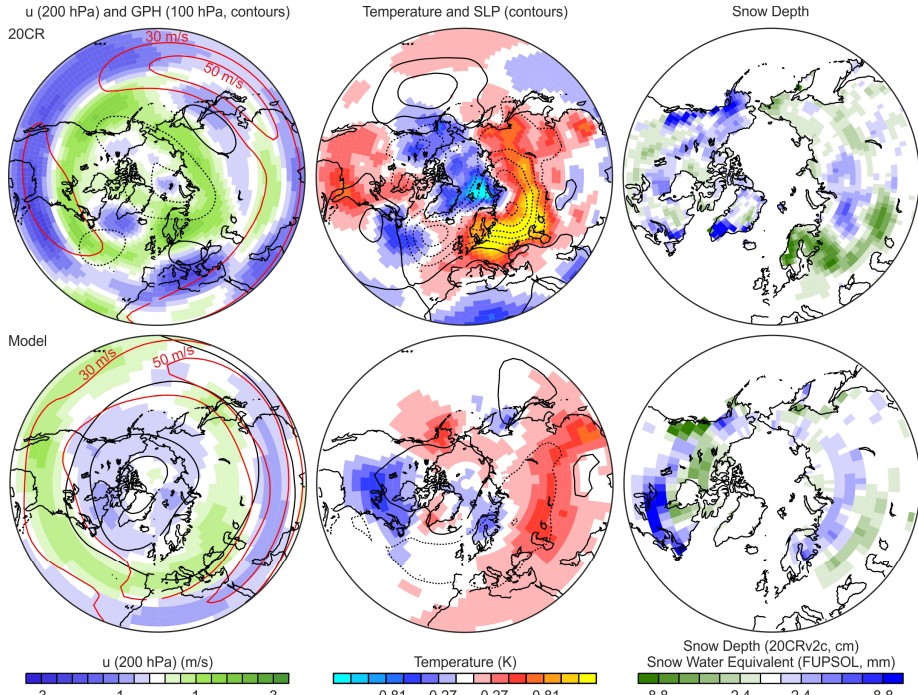

**Figure 3:** Composite fields for easterly minus westerly QBO phases for (left) 200 hPa zonal wind (shading, red contours indicate climatology) and 100 hPa GPH (black contours, spacing 24 gpm, symmetric around zero, dashed are negative),
5    (middle) surface air temperature (shading) and SLP (contours, spacing 0.6 hPa, symmetric around zero, dashed are negative) and (right) snow cover. See Fig. S3 for significance.



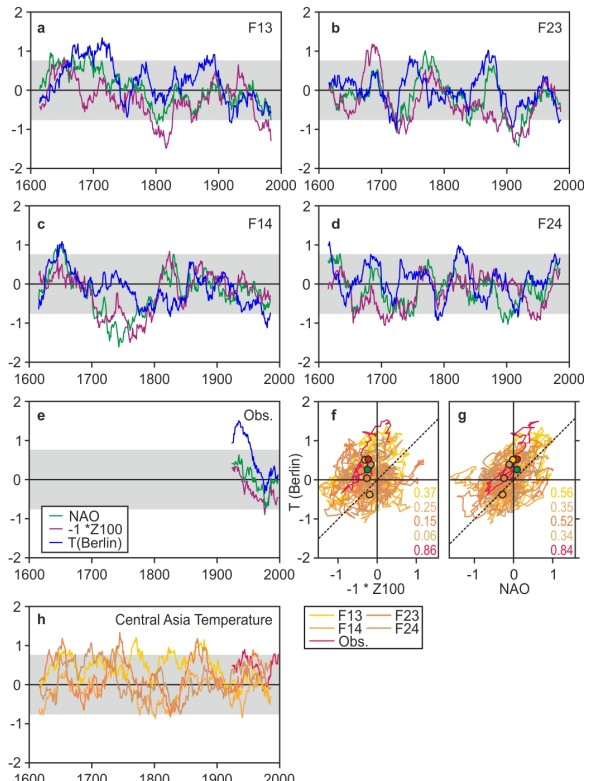

**Figure 4:** Standardized differences between easterly minus westerly QBO phases in 30-yr moving windows for NAO, Z100 and Berlin SAT in the four model simulations (a) F13, (b) F14, (c) F23, and (d) F24) and in (e) observation-based data (Z100 from reconstructions, NAO from 20CRv2c). Panels (f) and (g) show the same time series, but plotting Berlin SAT as a function of -1 * Z100 or NAO (filled circles indicate the standardized difference over the entire period and numbers give the correlations; the green circles indicates the standardized differences from ERA-Interim, 1979-2015, one-to-one lines are given in dashed). Panel (h) shows standardized differences between easterly minus westerly QBO phases in 30-yr moving windows for the Central Asia SAT index in HadCRUT4v and in the model simulations. Grey shading denotes an approximate 95% confidence interval for 30-yr averages.





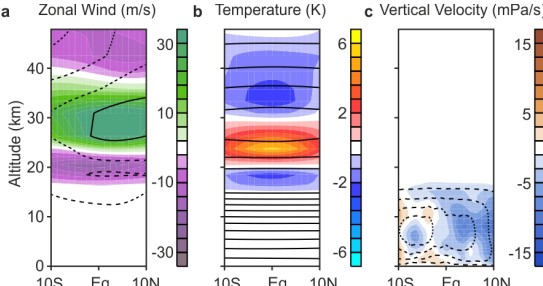

**Figure 5:** Composite fields (shading) and climatologies (contours) of ERA-Interim for boreal winter for easterly minus westerly QBO phases for zonal averages within 120-160° E for (a) zonal wind contours: (-50 to 50 m/s in steps of 20 m/s), (b) temperature (contours: 200 to 300 K in steps of 10 K) and (c) omega (-70 to 70 Pa/s in steps of 20 Pa/s) in Dec.-Feb. using the 70 hPa QBO definition.




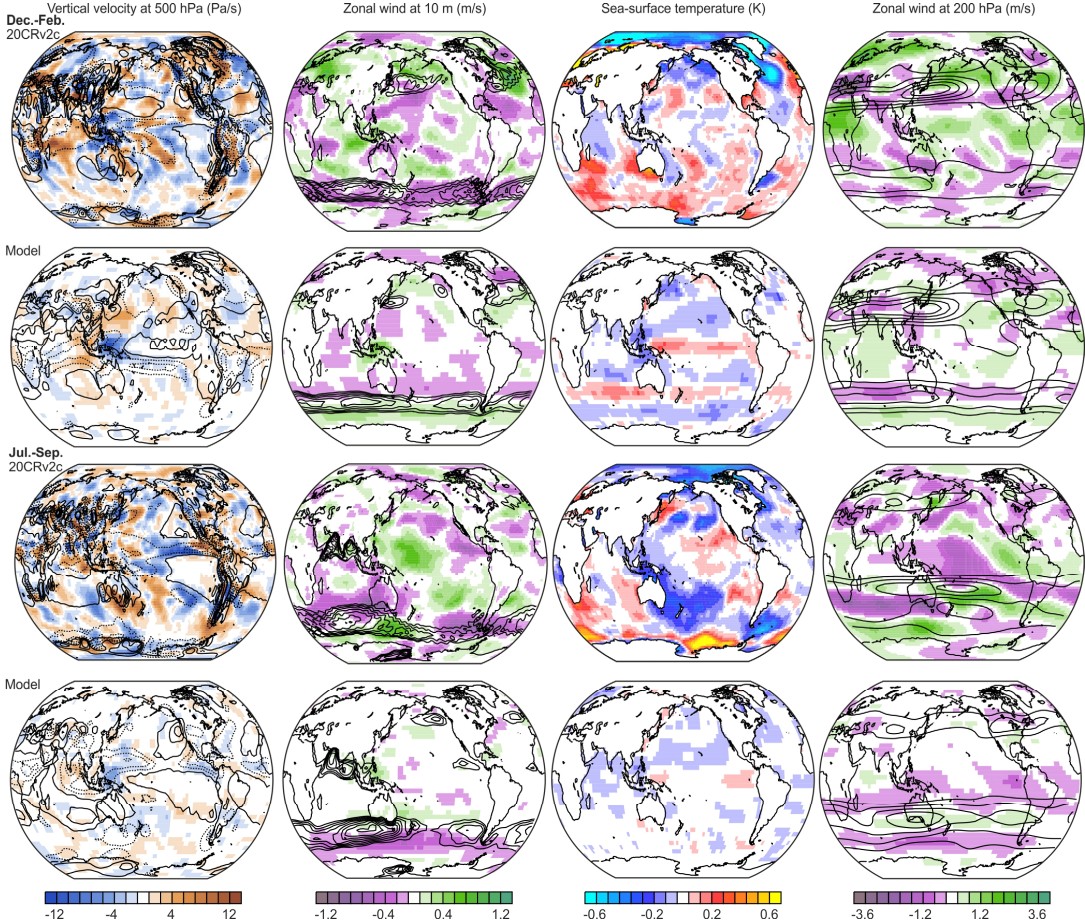

**Figure 6:** Composite fields for easterly minus westerly QBO phases for (left) 500 hPa vertical velocity (shading, contours indicate climatology, spacing is 40 Pa/s symmetric around zero, dashed are negative), 10 m zonal wind (shading, contours indicate climatology from 4 m/s in steps of 1 m/s), SST, and 200 hPa zonal wind (shading, contours indicate climatology from 20 m/s in steps of 10 m/s) for boreal summer and winter in 20CRv2c and in the model simulations. See Fig. S10 for significance.