# Peer review of "Multidecadal Variations of the Effects of the Quasi-Biennial"

_Atmospheric Chemistry and Physics, 2016_

## Referee Comment (RC1) · I. Roy (Referee) · 23 Jul 2016

Multidecadal Variations of the Effects of the Quasi-Biennial Oscillation on the Climate System- review by Indrani Roy

This study first investigates additional observations of stratospheric winds to test the existing QBO reconstruction. It then uses that time series to test reconstructions and reanalyses data. It also compares the results with outputs from coupled ocean-atmosphere-chemistry climate model with forced reconstructed QBO. This work mainly focuses on testing tropical-extratropical interaction in the stratosphere and subsequent downward propagation. It also investigates tropical deep convection, surface air temperature, precipitation, snow cover, tropospheric wind fields, and hurricane tracks. The QBO imprint on the northern polar vortex and Berlin surface air temperature both

shows different results during 1908-1957 to that from the later period. However, there is a small influence of the QBO on tropospheric climate in general. It is an interesting research and merits publication after minor revision.

Main Points:

 c Page 7, line 27: Discuss clearly the method 'heteroscedastic t-testing to assess statistical significance'. Also page 8, line 1: Explain standardized differences. As those are the main methods to capture signal QBO east to that from QBO west, describe those methods in details and give the formula.

 c Page 10, line 20: "predominantly negative, as expected from the Holton-Tan (H-T) effect." Also in page 9, line 7: there are some remarks about H-T effect. It is better to mention some discussions in this context relating to change in the behaviour of H-T effect during period 1977-1997. Lu et al. (2008) discussed that H-T effect weakened and even reversed around that time. Though it was statistically significant during solar min and 1959-1976 and 1998-2006 for extended winter (Nov-April).

Ref:

Lu, H., M. P. Baldwin, L. J. Gray, and M. J. Jarvis (2008), Decadal-scale changes in the effect of the QBO on the northern stratospheric polar vortex, J. Geophys. Res., 113, D10114, doi:10.1029/2007JD009647.

 c Page 7, line 29: It is good as the different period is used '1908-1957 and 1958-2012'. Considering longer period, Roy (2014) also showed that some tropospheric signatures were different before and after 1958. I liked the idea of Table 1. Give little discussions with a possible explanation for the change in the northern hemisphere polar vortex and Berlin surface air temperature during that periods. One explanation could be that there was a change in mean state of tropospheric circulation during 1958 (Vecchi and Soden (2007)). Change in tropospheric circulation can directly impact Berlin surface temperature. Also, the circulation change can modulate upward propagating

planetary wave activity and subsequently can influence Polar Vortex. Page 1, line 29: In the abstract it is mentioned that 'In the model simulations, likewise, both links tend to appear alternatingly, suggesting a more systematic modulation.' Discuss possible mechanism in the text, in line with the earlier paragraph, to make that argument strong.

Ref:

Roy, I., (2014), 'The role of the sun in atmosphere-ocean coupling' International Journal of Climatology, 34 (3), 655-677, doi:10.1002/joc.3713.

Vecchi GA, Soden BJ. 2007. Global warming and the weakening of the tropical circulation. Journal of Climate 20: 4316–4340.

• Fig. 4: It is hard to distinguish the colours in f, g and h.

Minor comments:

• Heading 3.1.2.: Apart from Berlin, other places also show interesting results as you discussed in this section. Hence better not to mention only Berlin in this heading and change it accordingly.

• Page 3, line 14: correct the inverted comma.

• Page 5, line 13: correct the inverted comma.

• Page 10, line 3: different period shows a different result. Also page 10, line 5: "The 30-yr moving windows difference in the model simulations shows a similar behaviour." It is an interesting result that model results match with observation. Elaborate those parts.

• Page 7, line 26: Explain a bit more why and mention roughly how many easterly and how many westerly?

• Page 7, line 11: replace 'season' with 'months'.

• Page 4, Line 18: Use the full form with abbreviation at the first place in all cases,

for e.g. CHUAN.

• Page 5, line 27, Heading 2.2: It is not only Troposphere as in line 17 it mentions Z100 (defined at 100 hPa level). Hence change the heading accordingly.

• Page 4, line 13: It is easier for readers if there is a mention Berson's profile 'that started on . . ..'

• Page 7, line 15: Instead of boreal summer (JJ) only mention June-July (JJ) and likewise for ND (Nov-Dec)?

• Line 17: change 'that season' with 'in those months'.

• Table 1 caption: 'F??' does not look nice and change it.

• Table 1: F13 and F14 instead F013 and F014.

• Table 1: Used abbreviated form of REC and NNR without defining in the text.

• Fig.1 caption: In the second line for dots mention about 'first three rows'. If possible improve dots in the first row, as some are placed below the x-axis.

• Fig 2 caption: Boreal winter is mentioned in the first line. But afterwards it is stated that for (c) GPH in Jan.-Mar., using the 50 hPa QBO early winter. Omit 'boreal' from the first line.

• Page 10, line 20: Describe this line a bit more.

• Page 10, line 15, 'The SAT signal over North America is totally different.' Any explanation?

• Page 24, line 8: use ';' instead of last ','.

• Fig S1: For 30th Aug 1908, two different colour profile indicates opposite. Give a possible explanation for that.

• Fig S3: Mention about the data in the figure caption.

• Fig S10: Make the significant region clearer. In the current form, it is hard to distinguish.

Please also note the supplement to this comment:
http://www.atmos-chem-phys-discuss.net/acp-2016-502/acp-2016-502-RC1-supplement.pdf
* * *

---

## Referee Comment (RC2) · Anonymous Referee #2 · 8 Aug 2016

Review on the manuscript "Multiple variations of the effects of the Quasi-Biennial Oscillation on the climate system" by Stefan et al.

The manuscript is motivated by the fact that the tropospheric impact of the Quasi-Biennial Oscillation is previously found to be weak, despite the dominance of the QBO in the equatorial stratosphere at interannual timescale, along with the known mechanisms by which the QBO influences the troposphere and the surface. The main objective of the study hence is to clarify the QBO connection to the surface by utilizing over 100 year long reconstructed and simulated data. The reconstructed data, which is mostly carried over from a previous study, goes back to 1908, and the QBO-forced model simulations extends even further back to 1600. Reanalysis data, such as ERA-20C and ERA-interim, is additionally used as well in the study. The analysis result primarily replies on the composites for two QBO phases.

Overall, it is appreciable that the study provides an attempt examining the question in long observational data and simulations. The composite analyses for QBO phases during chosen two sub-periods of the data conclude that the tropospheric impact of the QBO is small and appears intermittently. However, I feel that the conclusion could be more carefully drawn, as suggested below. In addition, despite the question itself is interesting, the manuscript is quite difficult to read, which limits the contribution of the study. For example, descriptions are missing for the use of data and the statistical significance test, while the historical background on observation is quite lengthy. Also, I find there are too many jumps back and forth to follow their analysis and figures. Lastly, figures and tables are not explained clearly.

Sec. 2: Although detailed information on data used in QBO reconstruction and tropospheric climate is described, it is very difficult to follow which one goes to which period and which exact period is used for the study. A table or schematics showing types of observations and period used in reconstruction will be certainly helpful to readers.

Model: Indicate the vertical resolution between 300hPa - 50 hPa, which would be important for the QBO to influence the tropical convection.

The composite is done for QBO phases. The phases are defined by the sign of the wind without the information on the amplitude. Has the amplitude been considered? For instance, sampling only when QBO index exceeds certain threshold may reduce discrepancy between the sub-periods and between the datasets.

l.14, p.2: It is puzzling that the paragraph starts that the impact of QBO on the troposphere is weak, but the main context of the paragraph supports the tropospheric signal of the QBO.

l.13, p.8: Figure 2 must be for 1979 and onward, and one of the purposes of the plot is to show the pattern of the H-T effect. But the results from other datasets often do not support the H-T mechanism. To better visualize why this is the case, I recommend to examine the patterns in other datasets as well, such as REC+NNR, 20CRv2c, and the

model simulations. This can tell use about the following: Z100 does not tell us about tropical response, which should be consistent with ERA-interim by construction. Extratropical Z100 response of REC+NNR for 1957-2014 is consistent with ERA-interim and the H-T effect. Some model simulations also support this picture. However, it is not the case for 20CRv2c.

l.19, p.8: Please indicate pressure levels in Fig. 2. Absence of this hinders comparison with Table 1. Statistical significance will be very useful.

Table 1: Please define REC+NNR. Also add period 1979-2014 and ERA-interim result; this period is shown in Figs. 2 and 5. Lastly, it is difficult to picture what is going on based on the mean pictures of the two sub-periods. Temporal variation would be certainly useful to understand each data. For example, running correlations between QBO index and Z100 may be able to tell us about the discrepancy between data. At least all the data are expected be consistent for the latest decades, say after 1979 and onward. One may be able to trace back where the discrepancy is started.

Figure 3: Another row using REC+NNR can be a useful reference as REC+NNR better compares with the simulations based on Table 1.

Model simulations in Fig. 4 provide a nice visualization that the impact of QBO is weak. However, some metrics can be useful to summarize the result. For example, what are the correlations among models? Also, the authors probably have tested different windows for the moving average. What is the reason of choosing 30 years? It should be tested to indicate robustness of their result to the choice of the window.

Figure 5: Statistical significance test especially for the vertical motion is required.

Figure 6: Fig. S10 shows just one row. Besides it appears there is almost no significant areas, which is not clear in the text. I feel that Figure 6 is misleading as if there is strong QBO signal.

---

## Author Response (AR1)

**Reply to the reviewer's comments**

*Reviewer 1:*

Many thanks for the helpful review and suggestions. We have improved the manuscript along the lines indicated. Please find below the replies.

**Comment:** Page 7, line 27: Discuss clearly the method 'heteroscedastic t-testing to assess statistical significance'. Also page 8, line 1: Explain standardized differences. As those are the main methods to capture signal QBO east to that from QBO west, describe those methods in details and give the formula.

**Reply:** We have added more detail on the method, including a formula and references for the testing.

**Comment:** Page 10, line 20: "predominantly negative, as expected from the Holton-Tan (H-T) effect." Also in page 9, line 7: there are some remarks about H-T effect. It is better to mention some discussions in this context relating to change in the behaviour of H-T effect during period 1977-1997. Lu et al. (2008) discussed that H-T effect weakened and even reversed around that time. Though it was statistically significant during solar min and 1959-1976 and 1998-2006 for extended winter (Nov-April).

**Reply:** We have rephrased the text and have added the reference.

**Comment:** Page 7, line 29: It is good as the different period is used '1908-1957 and 1958-2012'. Considering longer period, Roy (2014) also showed that some tropospheric signatures were different before and after 1958. I liked the idea of Table 1. Give little discussions with a possible explanation for the change in the northern hemisphere polar vortex and Berlin surface air temperature during that periods. One explanation could be that there was a change in mean state of tropospheric circulation during 1958 (Vecchi and Soden (2007)). Change in tropospheric circulation can directly impact Berlin surface temperature. Also, the circulation change can modulate upward propagating planetary wave activity and subsequently can influence Polar Vortex. Page 1, line 29: In the abstract it is mentioned that 'In the model simulations, likewise, both links tend to appear alternatingly, suggesting a more systematic modulation.' Discuss possible mechanism in the text, in line with the earlier paragraph, to make that argument strong.

**Reply:** We have added the references and also some text explaining the possible dependence on the background. We have no results to offer here, however. We tested several measures of the base state (e.g. circulation indices), but did not find agreement between models and observations. Moreover, given the time window used, the signals in the observations must be enormous to be significant. Therefore, although we found for instance a very strong correlation between the non-stationarity of the signal and the AMO in the observations, it is not significant in the observations alone and the same is not found in the model. It therefore remains a hypothesis. Another potential influence are decadal latitudinal shifts in the jet. We added a little bit of text, but no results.

**Comment:** It is hard to distinguish the colours in f, g and h.

**Reply:** Done

Minor comments. All minor comments were incorporated into the manuscript.

*Reviewer 2:*

Thank you very much for the review and the suggestion. We have improved the manuscript along the lines indicated. Specifically, we have improved significance testing (and its description), as outlined below and in the replies to the other reviewer. The use of data is now illustrated in a schematic, and the historical background is shortened. We have tried to straighten the argumentation at times.

**Comment 1:** Sec. 2: Although detailed information on data used in QBO reconstruction and tropospheric climate is described, it is very difficult to follow which one goes to which period and which exact period is used for the study. A table or schematics showing types of observations and period used in reconstruction will be certainly helpful to readers.

**Reply:** We have added a schematic figure (Fig. S3) that illustrates the data sets used, time periods, and analysis windows.

**Comment 2:** Model: Indicate the vertical resolution between 300hPa - 50 hPa, which would be important for the QBO to influence the tropical convection.

**Reply:** The vertical resolution around the tropopause is indicated.

**Comment 3:** The composite is done for QBO phases. The phases are defined by the sign of the wind without the information on the amplitude. Has the amplitude been considered? For instance, sampling only when QBO index exceeds certain threshold may reduce discrepancy between the sub-periods and between the datasets.

**Reply:** We require the winds to be above 5 m/s, otherwise QBO phase is classified as neutral. In this sense, the amplitude of the QBO is included to a small extent. This is made clear in the revised manuscript.

**Comment 4:** It is puzzling that the paragraph starts that the impact of QBO on the troposphere is weak, but the main context of the paragraph supports the tropospheric signal of the QBO.

**Reply:** We have rephrased the beginning of the paragraph.

**Comment 5:** Figure 2 must be for 1979 and onward, and one of the purposes of the plot is to show the pattern of the H-T effect. But the results from other datasets often do not support the H-T mechanism. To better visualize why this is the case, I recommend to examine the patterns in other datasets as well, such as REC+NNR, 20CRv2c, and the model simulations. This can tell use about the following: Z100 does not tell us about tropical response, which should be consistent with ERA-interim by construction. Extratropical Z100 response of REC+NNR for 1957-2014 is consistent with ERA-interim and the H-T effect. Some model simulations also support this picture. However, it is not the case for 20CRv2c.

**Reply:** Yes, the analysis is since 1979, and this is now clearly stated in the revised manuscript. We also add a figure where we show the same for 20CRv2c and ERA-20C for the exact same period (new Fig. S4). Some text has been added.

**Comment 6:** Please indicate pressure levels in Fig. 2. Absence of this hinders comparison with Table 1. Statistical significance will be very useful.

**Reply:** We added the pressure levels as well as statistical significance. (We also added the same to Fig. 5)

**Comment 7:** Table 1: Please define REC+NNR. Also add period 1979-2014 and ERA-interim result; this period is shown in Figs. 2 and 5. Lastly, it is difficult to picture what is going on based on the mean pictures of the two sub-periods. Temporal variation would be certainly useful to understand each data. For example, running correlations between QBO index and Z100 may be able to tell us about the discrepancy between data. At least all the data are expected be consistent for the latest decades, say after 1979 and onward. One may be able to trace back where the discrepancy is started.

**Reply:** We added ERA-Interim results. The analysis of moving window composites between the QBO and Z100 was repeated for all data sets (for consistency with the rest of the paper, we did not use correlations). Results show that in recent years, differences between the data sets are small, while they increase further back in time. We accounted for this in the revised manuscript by adding ERA-Interim as well as ERA-20C to Table 1 and adding a new period, 1979-2014 to Table 1. Some text is added on that.

**Comment 8:** Figure 3: Another row using REC+NNR can be a useful reference as REC+NNR better compares with the simulations based on Table 1.

**Reply:** We have performed that analysis but do not show it in the revised paper. The corresponding figure for the reconstructions alone is already shown in the supplement. Showing the figure with the reconstructions merged with a recent reanalysis would add another complexity since the field reconstructions must be extended with ERA-40 (to which it was calibrated), while the index reconstructions must be extended with NCEP/NCAR (they stem from two different papers). Therefore, two more data sets would have to be added and discussed. Showing only the period 1957, which we do in the supplement, is cleaner in this sense.

**Comment 9:** Model simulations in Fig. 4 provide a nice visualization that the impact of QBO is weak. However, some metrics can be useful to summarize the result. For example, what are the correlations among models? Also, the authors probably have tested different windows for the moving average. What is the reason of choosing 30 years? It should be tested to indicate robustness of their result to the choice of the window.

**Reply:** Concerning the correlation among models, the indices are nearly uncorrelated as correlation can only come from the forcings or the QBO. We added a sentence on that. Concerning the window length, a period of 30 years was chosen as this represents the typical time period of data availability in many studies. However, as the effects could be aliased by variability modes of a similar periodicity, we also tested other window lengths (20 to 50 years) and found the same results. A sentence on that is added.

**Comment 10:** Figure 5: Statistical significance test especially for the vertical motion is required.

**Reply:** Significance is added.

**Comment 11:** Figure 6: Fig. S10 shows just one row. Besides it appears there is almost no significant areas, which is not clear in the text. I feel that Figure 6 is misleading as if there is strong QBO signal.

Reply: Yes, almost nothing is significant, and this is important to note. We added a sentence to the text on that.

[revised manuscript text omitted]

---

## Referee Report (RR1)

**Multidecadal Variations of the Effects of the Quasi-Biennial Oscillation on the Climate System- 2nd review by Indrani Roy**

In the revised version, I find almost every point as raised by the referees are addressed. However, I am mentioning few very minor points. Hence I recommend for publication after a minor revision.

1. Fig. 4: Colours of f, g, h are not changed in the new version though mentioned that it is corrected. If it is difficult to incorporate, I am fine with that.
2. Page 9, Line 29: It is hard to read the formula. Make it clear.
3. Supplementary Fig. S1 Caption: correct spelling 'not' with 'note'.
4. Fig. S3 Caption: Mention about red dot and arrows, what they indicate.
5. Fig S10-S11 Captions: You could write in those Supplementary figure captions as ' same as Fig S9, but for Sea Surface Temperature', etc. It avoids repetitions.
6. Table 1: NAO why 0 after 1979, what does it imply?
7. Page 14, line 20: 'Changes in wave activity diagnostics are debated' - give reference of that work.
8. Table 1: $PWC_{JAS}$ during 1979-2015 gave significant results (same signed) for all three different datasets. Mention that interesting results.
9. Also for $PWC_{JAS,}$ all observations are –ve, irrespective of time periods. But model results are mostly positive. If you have any explanation, mention that.
10. Table 1: In observation, almost all the $PWC_{DJF}$ are with positive sign, though –ve for $PWC_{JAS.}$ Do you have any explanation for that?
11. For S3 if possible you could change the colour of significant level to red or any bright colour. It is difficult to recognize in current form.

---

## Author Response (AR2)

acp-2016-502
Multidecadal Variations of the Effects of the Quasi-Biennial Oscillation on the Climate System
Stefan Brönnimann, Abdul Malik, Alexander Stickler, Martin Wegmann, Christoph C. Raible, Stefan Muthers, Julien Anet, Eugene Rozanov, and Werner Schmutz

*Reply to the reviewer's comment*
1. Fig. 4: Colours of f, g, h are not changed in the new version though mentioned that it is corrected. If it is difficult to incorporate, I am fine with that.
The colours were in fact changed, but obviously not enough - now the changes are more clear.

2. Page 9, Line 29: It is hard to read the formula. Make it clear.
This is a PDF-conversion issue - I will have carefully check the proof.

3. Supplementary Fig. S1 Caption: correct spelling 'not' with 'note'.
Done.

4. Fig. S3 Caption: Mention about red dot and arrows, what they indicate.
Done.

5. Fig S10-S11 Captions: You could write in those Supplementary figure captions as ' same as Fig S9, but for Sea Surface Temperature', etc. It avoids repetitions.
Done (only Fig. 11, as Fig. S10 is slightly different, though).

6. Table 1: NAO why 0 after 1979, what does it imply?
The value is 0.0003, but since the table lists only 3 digits after the comma, it is 0.000, so this is correct.

7. Page 14, line 20: 'Changes in wave activity diagnostics are debated' - give reference of that work.
The sentence is rephrased.

8. Table 1: PWCJAS during 1979-2015 gave significant results (same signed) for all three different datasets. Mention that interesting results.
9. Also for PWCJAS, all observations are –ve, irrespective of time periods. But model results are mostly positive. If you have any explanation, mention that.
10. Table 1: In observation, almost all the PWCDJF are with positive sign, though –ve for PWCJAS. Do you have any explanation for that?
The three comments belong together. The reviewer is right that this was not emphasized enough, our discussion on this was ultra-short. In the revised text, this section is extended to: „We found a slight, not significant increase in observation-based data during boreal winter. In summer, we find a significantly negative response (weakening Walker circulation during easterlies) during the ERA-Interim period in all data sets (and a response of the same sign – though not significant - in all other subperiods and data sets). In contrast, in the model the Walker circulation is stronger for easterly than for westerly QBO in both seasons (DJF and JAS), in some simulations highly significant. The former is consistent with increased convection over the Pacific warm pool area and is consistent with observations (Fig. 5), whereas the latter is not consistent with increased convection and the sign is different from that found in the observations.“

11. For S3 if possible you could change the colour of significant level to red or any bright colour. It is difficult to recognize in current form.

Not sure which figure is meant. Fig. S3 (and Fig. 3) have no significance indicated, Fig. S5 canont be meant as red colour would completely vanish here.